# Personality Traits and Motives in Table Tennis Players

**DOI:** 10.3390/ijerph191710715

**Published:** 2022-08-28

**Authors:** Filippo Marchese, Ivan Malagoli Lanzoni, Patrizia Steca

**Affiliations:** 1Department of Psychology, University of Milan-Bicocca, 20126 Milan, Italy; 2Department of Biomedical and Neuromotor Sciences, University of Bologna, 40126 Bologna, Italy

**Keywords:** table tennis, Big Five personality factors, motives, self-determination theory

## Abstract

This study aims to investigate table tennis players’ personality traits and motives in the frame of the Big Five personality model and the self-determination theory (SDT) of motivation. A total of 447 Italian table tennis players ranging in level of play between the regional and international levels participated in the study. They completed a self-report questionnaire measuring their personality traits and motives to play table tennis. Findings showed conscientiousness as the most distinctive trait of table tennis players. No differences were detected between elite and non-elite players. Table tennis players are mainly motivated by factors belonging to the intrinsic pole of the self-determination motivational continuum. External reinforcements represent a minimal incentive to play this sport both for elite and non-elite athletes. The current findings help clarify the relationships between personality traits, playing certain types of sports, and achieving different performance levels. We conclude by outlining implications for applied sport psychology.

## 1. Introduction

The study of personality and motivation in the field of sport and exercise psychology focuses on answering several key questions: Do these factors influence the decision to take part in one type of sport rather than another? Do they matter in terms of involvement and commitment? Are there any associations between them and sport performance? Answering these questions may help clarify the role of psychological factors in playing and being successful in a given sport. Answers may also contribute to developing practical interventions aimed at orienting sport choices. The current study aims to investigate personality traits and motives in the population of table tennis players.

Most research on personality traits in sports psychology is framed in the Big Five model, which organizes personality based on the five traits of extraversion, agreeableness, conscientiousness, emotional stability, and openness [1]. Extraversion refers to a high level of energy accompanied by social behaviors and a general confidence in life situations. Conscientiousness expresses precision, organization, perseverance, responsibility, and motivation in achieving personal goals. Emotional stability identifies people who can avoid responding with negative emotions to circumstances of frustration, threat, or loss. Openness indicates curiosity about the ideas and values of others, as well as preference for the new. Findings have shown higher scores in all five factors in professional athletes compared to populations not involved in sports [2,3,4]. Significant differences have also been found among different types of athletes. Individual sports athletes have shown greater levels of conscientiousness, emotional stability, and openness, but lower extraversion, compared to athletes of team disciplines [3,5,6,7]. More recent studies partially confirmed these results, showing that individual-sport athletes are more open but also more energetic than team-sport athletes [4,8]. One previous study also showed significant differences between more successful and the less successful athletes, as the former reported higher agreeableness, conscientiousness, and emotional stability than the latter [4].

Overall results still remain rather heterogeneous and this is largely due to a number of methodological issues. Among them is the small sample size; studies in the literature hardly exceed 200 subjects and, in the worst cases, count just a few dozen. Finding an adequate number of participants may be difficult in a particular sport and this problem is often solved by bringing together different sports (e.g., [5,6]). However, this modus operandi carries with it the risk of making the sample very heterogeneous. As a consequence, differences among sports remain largely under-investigated and very little is known on traits most common to each sport or type of sport. As regards table tennis, Lopez and Santelices [9] showed conscientiousness as the most dominant personality trait in a sample of elite players.

Motivation has been extensively investigated in psychology applied to sports and exercise and no doubt is left as regards the pivotal role that motivational factors play in doing sports and reaching performance excellence [10,11].

Self-determination theory (SDT) [12] is one of the most recognized theoretical models in this field. SDT defines a motivational continuum ranging from the lack of motivation to intrinsic motivation, positing extrinsic motivation between the two.

Whereas extrinsic motivation refers to behaviors and commitment to activities with the goal of earning a reward or avoiding something unpleasant (e.g., a punishment), either external or self-inflicted, intrinsic motivation, on the other hand, describes activities that are considered rewarding for their own sake.

Some activities (for example, hobbies) are usually motivated more by intrinsic motivation than extrinsic, whereas the opposite is true for other activities (for example, work). That said, both types of motivation can support carrying out the same activities. We can play the piano, swim, read, and do many other things because of the pleasure and gratification we derive from doing them, or we may engage in them for a “benefit”: applause or a good grade. Only in the first case, however, does the person feel real interest, feel their abilities are at stake, and does not spare time and energy and place important personal goals in the activity. Being motivated by intrinsic motivation to play the piano, swim, and read is, therefore, more positive for people than doing the same activities because of external incentives and reinforcements [13,14].

Intrinsic motivation is particularly relevant in sports [15]. Athletes driven by intrinsic motives generally find interest in the task itself, experiencing the sport as a valid end as such and considering any external reinforcement as a secondary element [16,17,18]. Research on table tennis players has emphasized their high levels of intrinsic motivation. A recent cluster analysis work by Chu and colleagues [19] showed that self-determined was the predominant motivational profile for table tennis players. Similar findings were discovered by Martinent and colleagues [20] in an elite sample of table tennis players attending French federal sports centers. Despite the high pressure and the competitive environment, young athletes involved in the study were shown to be able to maintain high levels of intrinsic motivation. More specifically, a further study by Furjan-Mandić and colleagues [21] showed that the predominant motivational issues for table tennis players were fun and fitness.

Previous research has also demonstrated associations between personality and motives to do sport. In particular, conscientiousness, openness, and extraversion were found to be related to intrinsic motivation, whereas a low level of emotional stability was associated with extrinsic motivation [22,23,24].

Taken together, research on table tennis players’ personality and motivation is very thin and, to the best of our knowledge, no one has investigated the two realms simultaneously. Moreover, the sample sizes of previous studies are often very small (e.g., [9]) and the participants’ ethnicity and culture are homogeneous, thus limiting the generalizability of the findings. Therefore, the main aim of the present study was to investigate personality characteristics and motivational factors in table tennis players in order to expand the scientific knowledge on their psychological features.

According to the evidence supported by previous literature, the following hypotheses have been formulated: first, the personality profiles of table tennis players are expected to show higher levels of conscientiousness than other personality traits. Second, high-level athletes are expected to have higher levels of agreeableness, conscientiousness, and emotional stability than mid-level practitioners have. As regards the athletes’ most important reasons for playing table tennis, the hypothesis was that most motivations would be attributable to the intrinsic pole of the extrinsic–intrinsic continuum. Finally, we expected personality traits to be related to motives as reported in previous literature.

## 2. Materials and Methods

### 2.1. Participants and Categories

Four hundred and forty-seven competitive Italian table tennis players (age = 39 ± 15 years, mean value and SD; 399 males and 48 females) participated in the study. The level of the players was established by the official national ranking (www.fitet.org), updated at the beginning of the 2019/20 season. The ranking included six different categories, listed and described below:First category: from position 1 to position 12.Second category: from position 13 to position 120.Third category: from position 121 to position 452.Fourth category: from position 453 to position 2000.Fifth position: from position 2001 to position 5000.Sixth category: from position 5001 onwards.

Within the categories, the sample was distributed as follows: 11 in the first category, 60 in the second, 74 in the third, 112 in the fourth, 126 in the fifth, and 64 in the sixth. In order to improve the clarity of the results, a differentiation of the sample was performed between elite athletes (EA; *n* = 145; age = 30 ± 11 years, mean value and SD) belonging to the first three categories, and non-elite athletes (NEA; *n* = 302; age = 43 ± 15 years, mean value and SD) belonging to the fourth, fifth, and sixth categories.

Before filling out the questionnaire, participants received a brief presentation of the study containing information on its purpose, the amount of time it would take to fill out, the fact that participation was completely voluntary, and that the information collected would be confidential and used only for research purposes in full compliance with current privacy laws. In addition, exclusion criteria included minor athletes (under 18 years old). Moreover, their written informed consent to participate was required before the trials were carried out. Written information was also provided as to the study’s aims, the ensured anonymity of data collection and analysis, and the absence of associated risks.

### 2.2. Procedures

The procedure included administering a short survey to measure the variables of interest. Participants filled out an anonymous online questionnaire, submitted between October 2019 and March 2020. Only one response was allowed for each participant. Participants were recruited through the main communication channels table tennis players usually use, for example, a themed forum, newsletters, and an article published on the Italian Table Tennis Federation’s website.

### 2.3. Measures

Athletes were asked to answer some preliminary socio-demographic questions and indicate their level of play by category. Their personality was assessed through a list of 25 adjectives [25], which are among those most commonly used to describe personality in Italian and are well representative of each dimension of the Five Factors. *Resourceful* is a sample adjective for extraversion, *quiet* for emotional stability, *accurate* for conscientiousness, *innovative* for openness, and *selfless* for agreeableness. Each personality trait was measured by five adjectives evaluated on a response scale from 1 (“not at all”) to 5 (“completely”).

The motivational factors were measured through the *Motivation Participation Questionnaire* [26,27]. This instrument is made up of 25 items measuring the motivations that drive a person to do sports. The participants assigned a score to each item on a Likert scale ranging from 1 (“not important at all ”) to 7 (“extremely important”), based on how much the reason reported in the item represented a motivational component relevant to their playing table tennis. The instrument identified the following factors: *Skill development, Status, Friendship, Competition, Fun, Group-team exercise, Fitness, External reinforcements, Travel, Use of sports material*, and *Releasing energy and tension*. Validity and reliability of the two tests were established by previous studies for the Five Factors [25] and the Motivational Participation Questionnaire [28], respectively.

### 2.4. Statistical Analysis

Several statistical analyses were carried out on the collected data using the SPSS 25.0 software.

An ANOVA was used to investigate differences due to category (EA vs. NEA) in personality traits and motivational factors. In view of the large amount of evidence in the literature regarding the influence of age, especially about motivational factors [29,30,31], analyses were performed to control for its effect.

The Mann–Whitney test was used to analyze differences in the ranks of motivational factors between the elite and non-elite group.

Moreover, to analyze the relationships between personality traits and motivational factors, partial linear correlations were computed.

## 3. Results

### 3.1. Differences in the Big Five Traits between EA and NEA

Table 1 presents personality traits for EA and NEA from the highest to the lowest mean level. The two groups present the same order, characterized by conscientiousness as the leading personality trait.

In order to test personality differences between EA (elite athletes) and NEA (non-elite athletes), a series of ANOVAs was performed, keeping the effect of players’ ages under control. No significant differences were found between the two groups for Extraversion (F[1, 443] = 2.59; *p* = 0.11), Agreeableness (F[1, 443] = 0.19; *p* = 0.67), Conscientiousness (F[1, 443] = 0.85; *p* = 0.36), Emotional Stability (F[1, 443] = 0.03; *p* = 0.87) or Openness (F[1, 443] = 0.01; *p* = 0.98).

### 3.2. Motivational Factors in Table Tennis Players

In order to identify the main motivational factors influencing athletes’ choices to play table tennis, the mean value for each motivational factor was calculated and then the factors were ordered as shown in Table 2. The highest ranked factors were: *Fun* (M = 6.07, SD = 1.28), *Fitness* (M = 5.42, SD = 1.38), *Competition* (M = 5.23, SD = 1.28), and *Skill development* (M = 5.11, SD = 1.28).

### 3.3. Differences in Motivational Factors between EA (Elite Athletes) and NEA (Non-Elite Athletes)

Table 3 presents means of motivational factors, ordered from the highest to the lowest, separately for EA and NEA. The main difference is that Competition occupies a higher position for EA (second place), compared to mid-level athletes (fourth place). It is also interesting to note that the external reinforcements remain in last place for both categories of athletes, although with significantly different mean values (see Table 3).

Findings from the ANOVA aimed at comparing EA and NEA, controlling for age effect, showed significant differences for Status (F[1, 443] = 15.72, *p* < 0.001), External reinforcements (F[1, 443] = 10.49, *p* = 0.001), Travel (F[1, 443] = 16.67, *p* < 0.001), and Friendship (F[1. 443] = 12.00, *p* = 0.001). In particular, Status, External Reinforcements, and Travel were found to be at higher levels for EA. Otherwise, Friendship showed higher values in non-elite players.

The Mann–Whitney test was used to analyze motivational differences between EA and NEA in order to compare the two ranks. As shown in Table 4, EA presented significantly higher ranks for most of the motives (*Skill development, Status, Competition, Fun, Group-team exercise, Releasing energy, Use of sports material, Travel*).

### 3.4. Interrelationships among Personality Traits and Motivational Factors

Table 5 shows linear partial correlations between motivational factors and personality traits. *Extraversion* was the trait most correlated with motives for practicing table tennis. A positive moderate correlation was found between *Extraversion* and *Competition*. *Extraversion* also has weak positive correlations with *Skill development*, *Status*, *Fitness*, *External reinforcements*, *Use of sports material*, and *Travel*. *Conscientiousness* showed weak positive correlations with *Friendship*, *Group-team exercise*, and *Travel*.

## 4. Discussion

The present study aims to highlight the specificities of table tennis players in terms of personality traits and motivations to play. A greater level of personality traits and motives was detected when comparing athletes and non-athletes in previous studies [2,3]. The results of the present manuscript also confirm a greater level of personality traits and motives when comparing table tennis athletes and a normal population, respectively.

First of all, our study confirmed conscientiousness as the predominant personality trait in table tennis players, as previously reported by Lopez and Santelices [9] on a smaller group. Differently from our hypotheses, no differences were found between elite and mid-level players, with conscientiousness being the highest personality trait for both. This takes on an interesting meaning: it seems that the aspects that identify the “typical” personality required for a table tennis player do not emerge when the athlete excels in this discipline, as is true for other sports. Rather, it is as if they were a necessary condition to even begin playing table tennis. Indeed, this may be related to the first encounter with table tennis, which is usually difficult. When novices start to play table tennis, they may believe they already have a great deal of knowledge, strengthened by victories against acquaintances and challengers. They may seem convinced that they only need to learn some “trick of the trade” to make their game more effective. However, when they realize that their level of play is not as good as expected, they need to possess great perseverance and willpower to keep on playing. Furthermore, typical facets of conscientiousness are fundamental not only for novices, but also for experienced and established players. Precision, for example, is fundamental for improving athletes’ technical moves, which are very short and quick in table tennis. The work required to develop technical skills is inevitably based on small details of execution, which are very different from the work in sports such as tennis that require longer moves.

This goes hand in hand with motivational aspects, as shown by our findings. It is in fact plausible that the great demand for perseverance to keep playing table tennis ensures that the few who continue develop a strong intrinsic motivation and therefore a pleasure inherent in playing table tennis, thus transcending any external incentives.

Of course, given the status of table tennis in the world, the financial incentives (with a few exceptions) for athletes are very low, even at the elite level. This aspect is also highlighted by our results. In fact, the importance given to external reinforcements comes in last place in the ranking of the most important motivational aspects for both groups of players, although at significantly higher levels for EA. The almost total lack of importance given to external reinforcements has probably allowed the intrinsic motivation of table tennis players to remain unaffected by the well-known influence that external incentives have on motivation: activities that are initially intrinsically motivating move towards the extrinsic pole (e.g., [32]).

The study by Furjan-Mandić and colleagues [21], involving 138 subjects, pointed out that the predominant reasons for playing table tennis cited by their sample were to achieve status, to have fun, and to stay in good physical shape. In our sample, the factors of having fun and staying in good physical shape were the most relevant, occupying the first two places in the ranking drawn up according to the mean values in descending order. It is interesting to note that Furjan-Mandić and colleagues [21] traced the high score for staying in good physical shape to the specificity of their sample, university students in motor sciences who had chosen to take a table tennis course. They were therefore people who valued sports highly (regardless of the sport played) as a source of physical well-being, as they were motor sciences students. However, this result was also confirmed in our sample, which was not affected by these factors. In fact, our study does not only include university students in motor sciences, suggesting the importance given to the motivation of staying fit by playing table tennis itself and not from the academic background of the subjects.

Significant differences were also found between the elite and non-elite group. In particular, EA showed higher levels for Status and Travel motivations. This appears to be consistent, since it is normal for high-level practitioners to receive greater recognition for their skills and to have more opportunities to travel for national or international competitions. The group of mid-level players showed higher levels in the Friendship motivation. This also seems the natural consequence of how EA give more importance to issues such as competitiveness, thus leaving the more purely social aspect of sports in the background. Considering motivation, results showed higher ranks for most of the reasons behind playing sports for EA. This fact emphasizes that a great motivational boost is required to become an excellent player, and it must be deployed on all motivational factors, from intrinsic to extrinsic.

The highest motivation shown for table tennis players was Fun. There is therefore confirmation of the hypothesis that table tennis players show particularly high levels of intrinsic motivation. These hypotheses were derived from the work of Abuhamdeh and Csikszentmihalyi [33,34], who stressed the role of a competitive environment and of activities with uncertain outcomes. Both characteristics combine very well in the sport of table tennis.

As regards uncertain outcomes, the fact that a small detail is enough to determine the effectiveness or ineffectiveness of a technical move (as pointed out above, the movements are short, so a minimum variation is enough to affect the execution of a shot) means that the performance of a table tennis player balances on a very thin wire. This aspect, combined with the characteristically short score, makes table tennis a sport with a highly uncertain outcome, in which it is not uncommon to see strongly unpredictable results.

As far as competitiveness is concerned, table tennis players operate in a context in which competition is crucial. Typically, in individual sports one’s own frustration with defeat cannot be veiled by team performance; there is no shared responsibility. As a result, almost all the responsibility lies with the individual player, who is therefore more encouraged to develop high competitiveness. In addition, table tennis players’ performance is always inevitably related to that of others; it seems almost impossible to give an evaluation of one’s performance free from that of the opponent and therefore from the final result. This does not apply to all individual sports. In swimming or athletics, for example, the objective measurement of the performance times is available, which makes it easier to give a positive assessment independently of the performance of opponents. Table tennis players have no similar type of objective indicator; their performance will always be bound to the result they achieve by competing against another opponent. Therefore, implementing competitiveness also seems appropriate for this group.

Relationships among traits and motives to play table tennis partly confirmed previous findings [22,23,24]. Extraversion, because of its nature, is strongly related to high activity, which is usually linked to being prone to and motivated by action. This emerged as the trait most related to motivational factors. In particular, the highest correlation was found between extraversion and competition, probably due to the dominance trait being a central facet of extraversion. However, most of correlations are low and medium, so the impact of these results has to be considered in light of this.

In light of our findings, we suggest some useful application insights for the training of table tennis players.

First, with regard to personality, in light of our findings, children should be encouraged from an early age to develop behaviors related to the trait of conscientiousness. It may therefore be useful to motivate children to prepare their own equipment bags (ability to organize), to store the material used for the training in an autonomous, methodical, and orderly way. Moreover, it might be useful to offer them activities in parallel to table tennis, which can stimulate precision and perseverance, such as solving brain teasers or puzzles. More specifically, it could be useful to encourage children to carry out some targeted exercises for their conscientiousness, for example, for them to engage in their daily life observing details of how people are dressed. Otherwise, when they are faced with a problem, they can try to practice finding five or six different solutions for it; in fact, conscientiousness is positively correlated with problem solving ability [35], and this task allows them to stimulate it.

Second, our findings suggest reflecting on the potential of this sport to generate a strong intrinsic motivational component, once the initial difficulties are overcome. Consequentially, in accordance with goal setting theory [36], it is essential that athletes are clear about goals they want to achieve and for which of them they train every day. A sport psychologist’s support can facilitate this work and he or she can also help athletes to maintain a positive attitude when they are faced with difficulties or obstacles.

In addition, the two questionnaires used in the present investigation could be useful for a talent identification process. Indeed, personal traits and motives of beginners and young athletes can be compared with elite athletes and this can help to identify which young athletes have the best chance of becoming professional players someday.

Despite its merits, some limitations of the study should be taken into account. First, the data were collected through anonymous self-report tests, yet there may be a certain amount of distortion if the subjects wished to present themselves as better than they really are. Further limitations have been found in the difference in sample size between male and female players. There are far fewer female players at the national level than male players and this has led to an imbalance in the sample. Given the known gender differences in terms of personality in the literature over the years [37,38], different results may have emerged had there been a bigger female sample available. In addition, the kind of study conducted does not allow us to draw cause–effect conclusions in relation to the variables analyzed, but rather associative relationships.

## 5. Conclusions

The present study has provided interesting ideas for reflection on a sport as poorly known as table tennis. This study has therefore certainly contributed to the accumulation of results in this regard, which can be integrated with previous research, the findings of which have been almost confirmed. The sample size, which to our knowledge is greater than any other study on the psychology of table tennis players, was certainly a relevant aspect of our contribution that greatly adds to the current literature. In addition, this is the first study to deal in detail with the themes of the personality and motivation of table tennis players on an Italian sample.

As for personality traits, we confirmed conscientiousness as the most prominent feature of the personality of table tennis athletes. This identifies table tennis players as people who are driven to precision, organization, and perseverance in achieving their goals.

With reference to motivational issues, our results have highlighted the predominance of intrinsic motivational factors in table tennis players, perhaps due to the combination of competitive aspects and the uncertain outcome of table tennis matches. Fun was found to be the most important motivational theme, and this indicates that table tennis players know how to draw gratification in a particularly meaningful way from the sport itself. Future research should further shed light on the positive effects of table tennis across ages.

## Figures and Tables

**Table 1 ijerph-19-10715-t001:** Personality traits of table tennis players for elite and non-elite athletes.

EA	Mean ± SD	NEA	Mean ± SD
Conscientiousness	4.15 ± 0.67	Conscientiousness	4.07 ± 0.61
Agreeableness	3.94 ± 0.71	Agreeableness	3.96 ± 0.68
Extraversion	3.84 ± 0.67	Extraversion	3.59 ± 0.70
Openness	3.45 ± 0.76	Openness	3.50 ± 0.78
Emotional Stability	3.41± 0.78	Emotional Stability	3.28 ± 0.82

**Table 2 ijerph-19-10715-t002:** Mean values in descending order of athletes’ reasons for playing table tennis.

	Mean ± SD
Fun	6.07 ± 1.28
Fitness	5.42 ± 1.38
Competition	5.22 ± 1.28
Skill development	5.11 ± 1.28
Releasing energy	4.72 ± 1.58
Group-team exercise	4.30 ± 1.51
Friendship	4.17 ± 1.95
Travel	3.55 ± 2.32
Status	3.53 ± 1.78
Use of sports material	3.21 ± 2.27
External reinforcements	2.12 ± 1.60

**Table 3 ijerph-19-10715-t003:** The most important motivations for playing table tennis in elite and mid-level athletes.

EA	Mean ± SD	NEA	Mean ± SD
Fun	5.99 ± 1.42	Fun	6.12 ± 1.21
Competition	5.79 ± 0.99	Fitness	5.41 ± 1.30
Fitness	5.43 ± 1.54	Skill development	5.01 ± 1.25
Skill development	5.18 ± 1.32	Competition	4.95 ± 1.32
Releasing energy	4.76 ± 1.67	Releasing energy	4.70 ± 1.54
Travel	4.70 ± 2.15	Group-team exercise	4.20 ± 1.51
Group-team exercise	4.52 ± 1.50	Friendship	4.06 ± 1.90
Status	4.48 ± 1.53	Use of sports material	3.19 ± 1.17
Friendship	4.39 ± 2.04	Status	3.01 ± 1.71
Use of sports material	3.25 ± 2.47	Travel	3.00 ± 2.19
External reinforcements	2.69 ± 1.65	External reinforcements	1.85 ± 1.50

**Table 4 ijerph-19-10715-t004:** Mann–Whitney test for each motivational factor between EA and NEA.

	Mann–Whitney	Z	p
Skill development	17,974	−3.094	0.002
Status	15,777	−4.834	<0.001
Friendship	20,905	−0.779	0.44
Competition	17,169	−3.730	0.08
Fun	19,697	−2.038	0.04
Group-team exercise	19,205	−2.120	0.03
Fitness	19,698	−1.751	0.08
External reinforcements	19,739	−1.754	0.08
Releasing energy	17,398	−3.539	<0.001
Use of sports material	17,797	−3.231	0.001
Travel	13,789	−6.384	<0.001

**Table 5 ijerph-19-10715-t005:** Partial correlations among motivational factors and personality traits.

EA	Extraversion	Agreeableness	Conscientiousness	Emotional Stability	Openness
Skill development	0.246; *p* < 0.001	0.066; *p* = 0.17	0.133; *p* = 0.01	−0.072; *p* = 0.13	−0.070; *p* = 0.14
Status	0.274; *p* < 0.001	−0.038; *p* = 0.43	0.136; *p* = 0.01	−0.048; *p* = 0.32	−0.001; *p* = 0.98
Friendship	0.113; *p* < 0.001	−0.043; *p* = 0.37	0.220; *p* < 0.001	0.061; *p* = 0.20	−0.039; *p* = 0.41
Competition	0.412; *p* < 0.001	−0.032; *p* = 0.51	0.077; *p* = 0.11	0.018; *p* = 0.71	−0.053; *p* = 0.26
Fun	−0.065; *p* = 0.17	−0.046; *p* = 0.33	0.083; *p* = 0.08	0.133; *p* = 0.01	0.039; *p* = 0.41
Group-team exercise	0.149; *p* < 0.001	−0.013; *p* = 0.79	0.239; *p* < 0.001	0.060; *p* = 0.21	−0.006; *p* = 0.89
Fitness	0.206; *p* < 0.001	0.041; *p* = 0.39	0.122; *p* = 0.01	0.027; *p* = 0.57	−0.004; *p* = 0.93
External reinforcements	0.267; *p* < 0.001	−0.050; *p* = 0.30	0.154; *p* < 0.001	−0.027; *p* = 0.57	0.019; *p* = 0.69
Releasing energy	0.174; *p* < 0.001	−0.009; *p* = 0.85	0.150; *p* = 0.01	0.028; *p* = 0.56	−0.012; *p* = 0.80
Use of sports material	0.208; *p* < 0.001	0.015; *p* = 0.76	0.112; *p* = 0.01	0.059; *p* = 0.22	0.071; *p* = 0.14
Travel	0.203; *p* < 0.001	−0.087; *p* = 0.07	0.210; *p* < 0.001	0.017; *p* = 0.72	0.004; *p* = 0.94

## Data Availability

Data in the form of anonymized questionnaires are available upon request from the corresponding author.

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
