# Peer review of "Personality Traits and Motives in Table Tennis Players"

_ijerph, 2022, doi:10.3390/ijerph191710715_

Round 1

Reviewer 1 Report

This article does not have interesting and new practical findings.
A lot of similar work has been done depending on whether it is a team or individual sport.
Research methodology and tools are not innovative.
Practical suggestions have not been presented and the results of this research have been more of an answer to the researcher's mental question than an answer to an important issue.
Good luck

Author Response

The Authors would like to thank the Reviewer for the constructive and precious advice. We found the comments very helpful to improve the paper. We have attached a copy of the Reviewers’ comments below and inserted our responses, outlining how each point has been addressed. The manuscript is revised with all changes clearly marked in red.

This article does not have interesting and new practical findings.

RESPONSE: we thank the Reviewer for every valuable comment. At the best of our knowledges, it is the first paper that investigates both, Personality Traits and Motives in Table Tennis Players with a large sample size. Moreover, we revised the manuscript accordingly, including a paragraph about practical suggestions in the field of table tennis (line 317-326).

A lot of similar work has been done depending on whether it is a team or individual sport.

RESPONSE: we thank the Reviewer for his/her comment. As suggested, the Introduction has been modified including more references about the differences between Team and Individual sports, respectively (references: 3,4,5,7, and 8) (Line 40-45).

Research methodology and tools are not innovative.

RESPONSE: the main aim of the present study was to investigate personality characteristics and motivational factors in table tennis players in order to expand the scientific knowledge on their psychological features. Therefore, we used the Methods suggested by previous literature about: Procedures, Measures, and Statistical Analysis.

Practical suggestions have not been presented and the results of this research have been more of an answer to the researcher's mental question than an answer to an important issue.

RESPONSE: We revised the manuscript accordingly, including the following part to the Discussion from Line 317 to Line 326 “In light of our findings, we suggest some useful application insights for the training of table tennis players. First, with regard to personality, children should be encouraged from an early age to develop behaviors related to the trait of conscientiousness. It may therefore be useful to motivate children to prepare their own equipment bags (ability to organize), to store the material used for the training in an autonomous, methodical, and orderly way. Moreover, it might be useful to offer them activities in parallel to table tennis, which can stimulate precision and perseverance such as solving brain teasers or puzzles. Second, our findings suggest reflecting on the potential of this sport to generate a strong intrinsic motivational component, once the initial difficulties are overcome”.

Good luck

RESPONSE: We would like to take this opportunity to express our sincere thanks to the Reviewer who identified areas of our manuscript in need of corrections or modifications.

Reviewer 2 Report

Dear authors,

thank you for this interesting study. It is well done, but there are a few comments that should be followed.

2.1 Participants: You have collected sociodemografic data. Please present them in the description of the participants. Could these perhaps have an impact on the results? Also, it would be interesting to know how long the participants have been playing table tennis and how long they have been in the current category, this could also affect the results.

2.2 Procedures: How were the participants recruited? Through the national tabble tennis federation? Please add this information.

2.3 measures: How good are these tests? Please provide information on the psychometric properties (reliability and validity aspects)

3 Results: How do the results compare to the normative values of the test procedures?

Table 5: The correlations are all significant because of the large sample. Overall, correlations are low to medium The highest coefficient for extraversion is .41, which cannot be called strong. Please elaborate on this in the discussion as well.

Author Response

The Authors would like to thank the Reviewer for the constructive and precious advice. We found the comments very helpful to improve the paper. We have attached a copy of the Reviewers’ comments below and inserted our responses, outlining how each point has been addressed. The manuscript is revised with all changes clearly marked in red.

Dear authors,

thank you for this interesting study. It is well done, but there are a few comments that should be followed.

2.1 Participants: You have collected sociodemografic data. Please present them in the description of the participants.

RESPONSE: we thank the Reviewer for his/her comment. This was added as proposed. We revised the manuscript accordingly, including sociodemographic data in the description of the participants (nationality, age, and sex), in Line 111-112.

Could these perhaps have an impact on the results?

RESPONSE: we thank the Reviewer for every valuable comment. The impact on the results of the sociodemographic data can not be totally excluded. Indeed, the 447 participants have the same nationality. Moreover, previous literature confirmed and supported our results about conscientiousness and motivational factors for participants of the same nationality (Lopez and Santelices [9], Furjan Mandic et al. [21]).

The age of the subjects can influence the results and it is well known in the literature [28-30]. Therefore “An ANOVA was used to investigate differences due to category (EA vs. NEA) in personality traits and motivational factors. In view of the large amount of evidence in the literature regarding the influence of age, especially about motivational factors [28-30], analyses were performed to control for its effect”. It has been included from Line 165 to Line 167.

Finally, an impact on the results can be connected to the sex of the participants. The number of women players in Italy is very limited, unfortunately. Therefore, it is not possible to compare the two groups. This variable was accordingly included in the paragraph about the limits of the study (Line 330-334)

Also, it would be interesting to know how long the participants have been playing table tennis and how long they have been in the current category, this could also affect the results.

RESPONSE: we thank the Reviewer for his/her comment. Due to the rules of the National Federation, it is possible to be included in the ranking only if a minimum of six official matches a year have been individually played. Moreover, the players can be downgraded if they don’t play for two following seasons. It shows the ranking is quite reliable and stable

2.2 Procedures: How were the participants recruited? Through the national tabble tennis federation? Please add this information.

RESPONSE: We revised the manuscript accordingly. This was added as proposed as supplementary information: “Participants were recruited through main communication channels table tennis players usually use, for example themed forum, newsletters and an article published on the Italian Table Tennis Federation’s website”. Line 138-141.

2.3 measures: How good are these tests? Please provide information on the psychometric properties (reliability and validity aspects)

RESPONSE: we thank the Reviewer for his/her comment. This was added as proposed as supplementary information “Validity and reliability of the two tests were established by previous studies for the Five Factors [25] and the Motivational Participation Questionnaire [28], respectively”.

The two cited papers were also included in the References:

  1. Barbaranelli, C., Caprara, G. V., Vecchione, M., & Fraley, C. R. (2007). Voters’ personality traits in presidential elections. Personality and Individual Differences, 42(7), 1199-1208. https://doi.org/10.1016/j.paid.2006.09.029
  2. Guedes, D. P., & Silvério Netto, J. E. (2013). Participation Motivation Questionnaire: tradução e validação para uso em atletas-jovens brasileiros. Revista Brasileira de Educação Física e Esporte, 27, 137-148. https://doi.org/10.1590/S1807-55092013005000003

3 Results: How do the results compare to the normative values of the test procedures?

RESPONSE: we thank the Reviewer for his/her comment. Accordingly, the following sentence has been included at the beginning of the Discussion “A greater level of personality traits and motives were detected comparing athletes and no-athletes by previous studies [2, 3]. The results of the present manuscript also confirm a greater level of personality traits and motives comparing table tennis athletes and normal population, respectively”.

Table 5: The correlations are all significant because of the large sample. Overall, correlations are low to medium The highest coefficient for extraversion is .41, which cannot be called strong. Please elaborate on this in the discussion as well.

RESPONSE: we thank the Reviewer for his/her suggestion. Accordingly, we included the following sentence in the Discussion “. However, most of correlations are low and medium, so the impact of these results has to be considered in sight of this”.

Reviewer 3 Report

The article addresses a very specific topic and presents a new approach that complements existing data on the topic. The sample size and study design add scientific value.

It would be important to have more recent bibliography for the introduction and foundation of the theme. The presentation of the references in line 43 needs reformulation.

I propose that the authorization of the ethics committee be mentioned directly in the procedures.

It will be necessary to inform about the measures to avoid more than one response from each participant (or if it is one of the limitations of the study).

Author Response

The article addresses a very specific topic and presents a new approach that complements existing data on the topic. The sample size and study design add scientific value.

The Authors would like to thank the Reviewer for the constructive and precious advice. We found the comments very helpful to improve the paper. We have attached a copy of the Reviewers’ comments below and inserted our responses, outlining how each point has been addressed. The manuscript is revised with all changes clearly marked in red.

It would be important to have more recent bibliography for the introduction and foundation of the theme.

RESPONSE: we thank the Reviewer for his/her comment. Due to the topic of the present investigation, we have already included the most recent articles from 2018 (Chu, T. L., Zhang, T., & Hung, T. M., 2018) and 2021 (Piepiora, P., 2021), respectively. Moreover, at the best of our knowledges, there is a very limited number of studies about Personality Traits and Motives in the field of Table Tennis, unfortunately.

The presentation of the references in line 43 needs reformulation.

RESPONSE: it has been done

I propose that the authorization of the ethics committee be mentioned directly in the procedures.

RESPONSE: we thank the Reviewer for his/her comment. As suggested, the following part has been added from Line 131 to Line 134 “In addition, exclusion criteria included minors players (under 18 years old). Moreover, their written informed consent to participate was required before the trials were carried out. Written information was also provided as to the study’s aims, the ensured anonymity of data collection and analysis and the absence of associated risks”.

It will be necessary to inform about the measures to avoid more than one response from each participant (or if it is one of the limitations of the study).

RESPONSE: We thank the Reviewer to have pointed this out. The following sentence has been added to Line 138 “Only one response was allowed for each participant”

Round 2

Reviewer 1 Report

The article can be accepted by strengthening the practical suggestions section.

Author Response

REVIEWER 1: The article can be accepted by strengthening the practical suggestions section.

RESPONSE: We would like to take this opportunity to express our sincere thanks to the reviewer 1 who identified area of our manuscript in need of corrections or modifications. Therefore, the practical suggestions section has been considerably improved including also specific references [35, 36].

[35] D’Zurilla, T. J., Maydeu-Olivares, A., & Gallardo-Pujol, D. (2011). Predicting social problem solving using personality traits. Personality and individual Differences, 50(2), 142-147. 10.1016/j.paid.2010.09.015

[36] Locke, E. A., & Latham, G. P. (1990). A theory of goal setting & task performance. Prentice-Hall, Inc.

We have rewritten and improved this section: “In light of our findings, we suggest some useful application insights for the training of table tennis players.

First, with regard to personality, in light of our findings children should be encouraged from an early age to develop behaviors related to the trait of conscientiousness. It may therefore be useful to motivate children to prepare their own equipment bags (ability to organize), to store the material used for the training in an autonomous, methodical, and orderly way. Moreover, it might be useful to offer them activities in parallel to table tennis, which can stimulate precision and perseverance such as solving brain teasers or puzzles. More specifically, it could be useful to encourage children to do some targeted exercises for their conscientiousness, for example get them to engage in their daily life observing details of how people are dressed. Otherwise, when they are faced with a problem they can try to practice finding five or six different solutions for it, in fact conscientiousness is positive correlated with problem solving ability [35], and this task allows to stimulate it.

Second, our findings suggest reflecting on the potential of this sport to generate a strong intrinsic motivational component, once the initial difficulties are overcome. Consequentially, in accordance with goal setting theory [36], it is essential athletes are clear about goals he wants to achieve and for which of them they train every day. A sport psychologist’s support can facilitate this work and he or she can also help athletes to maintain a positive attitude when they are faced with difficulties or obstacles.

In addition, the two questionnaires used in the present investigation could be useful for a talent identification process. Indeed, personal traits and motives of beginners and young athletes can be compared with elite athletes and this can help to identify which young athletes have the best chance of becoming professional players someday”.